# Localized Modes and Photonic Band Gap Sensitivities with 1D Fibonacci Quasi-Crystals Filled with Sinusoidal Modulated Plasma

Naim Ben Ali [1,2,*] , Youssef Trabelsi [2,3], Haitham Alsaif [4] , Omar Kahouli [5] and Zied Elleuch [6]

1   Department of Industrial Engineering, College of Engineering, University of Ha'il,
    Ha'il City 81451, Saudi Arabia
2   Photovoltaic and Semiconductor Materials Laboratory, National Engineering School of Tunis,
    University of Tunis El Manar, Tunis 1002, Tunisia; ytrabelsi@kku.edu.sa
3   Physics Department, College of Arts and Sciences in Muhail Asir, King Khalid University,
    Abha 61421, Saudi Arabia
4   Department of Electrical Engineering, College of Engineering, University of Ha'il,
    Ha'il City 81451, Saudi Arabia; h.alsaif@uoh.edu.sa
5   Department of Electronics Engineering, Applied College, University of Ha'il, Ha'il City 81451, Saudi Arabia;
    a.kahouli@uoh.edu.sa
6   Department of Computer Science, Applied College, University of Ha'il, Ha'il City 81451, Saudi Arabia;
    zi.elleuch@uoh.edu.sa
*   Correspondence: na.benali@uoh.edu.sa

**Abstract:** Using the transfer matrix method, the electromagnetic responses of 1D deformed and non-deformed quasi-periodic photonic crystals arranged in accordance with the Fibonacci sequence are theoretically studied. The gallium selenide (GeSe) and plasma materials (that is, electron density) are used to construct the multilayer Fibonacci structures. If this study is experimentally validated in the future, we intend to insert two transparent polymer film materials at the top and bottom of the structure, which are intended to protect the plasma material and prevent it from escaping and spreading outside the structure. The effect of the order of the Fibonacci sequence, the deformation of the thickness of the layers using a mathematical rule and the role of the plasma material in the reflectance response are discussed. We notice that the position and the width of photonic band gaps are sensitive to the Fibonacci sequence, the thickness and the density of the plasma material layers. In addition, the intensity of the resonance peaks can be controlled by adjusting the plasma material properties. The width of the photonic band gaps can be notably enlarged by applying a structural deformation along the stacks. The proposed structures have potential applications in tunable filters, micro-cavities for LASER equipment, which allow us to obtain an intense laser, and they are very useful in the communication field.

**Keywords:** Fibonacci sequence; plasma; localization; photonic; defect; resonance mode; tunability

## 1. Introduction

The photonic quasi-crystals built according to the deterministic aperiodic sequences have received particular attention in solid-state photonics [1–4]. These types of structures are made of building blocks, following the simple substitution rules of quasi-periodic mathematic sequences. They still show sharp diffraction patterns with a lack of translational symmetry and rotational symmetry, where the Fourier transformation of their geometrical structures in real space is defined by $F(k) = \lim_{N \to \infty} \sum R_N \exp(ik R_N)$, with $R_N$ as the atomic positions [5].

In the past decade, the use of plasma materials in photonic crystals (PCs), instead of the usual dielectric or metal materials, has attracted much attention due to their interesting properties compared to conventional photonic crystals [6–8]. The photonic crystals

involving plasma are made of alternating thin non-magnetized or magnetized plasmas and dielectric layers or metals and are characterized by a dispersive complex refractive index [9].

In recent decades, quasi-photonic multilayer structures built using incorporated super-conductors [10–12] or liquid crystals [13,14] or semiconductors [15] have received special attention and have been proposed for various applications, such as temperature sensing, pressure sensing [16], optical signal filtering [11,17] and light transport and localization [18]. Photonic quasi-crystals are non-periodic modulations of two or more refractive indices generated by recursive sequences, such as the Rudin–Shapiro [19], Thue–Morse [20] and Fibonacci [21] sequences. Quasi-periodic crystals (sometimes called aperiodic crystals) lie between the periodic and chaotic structures and still exhibit attraction, according to more recent studies [17,18]. The most common quasi-periodic sequence is the Fibonacci [21] one due to its incommensurable periods in the spatial spectrum of the structure—the so-called pure point spectrum. It shows Bragg-like peaks in its spatial spectrum [22]. The well-known symmetry of the Fibonacci sequence is generated by the inflation rule, $F_{k+1} = \{F_k, F_{k-1}\}$. The tuning of photonic band gaps (PBGs) through Fibonacci quasi-periodic structures containing a variety of materials, including semiconductor materials, has been investigated [23–25]. The localized modes in photonic quasi-crystals are generated by lattice vacancies breaking off the quasi-periodic symmetry. The Eigen frequencies of such localized modes are investigated using a super cell approximation in the plane wave method [26]. The Eigen frequencies and electromagnetic field profiles of multi-resonator photonic circuits are related to micro-cavity designs. These characteristics of quasi-photonic systems render these structures suitable for various applications with tunable logical elements, such as switches, modulators and filters [11,20], and they can be useful in optics and optoelectronic circuits.

In this work, 1D Fibonacci quasi-crystals filled with sinusoidal modulated plasma material are designed and theoretically studied. Here, using the transfer matrix method, we aim to study the effect of the plasma material, as well as the geometric thickness of layers and the quasi-periodicity of the structure on wave transmission, localization and filtering. The multi-output beam intensity and the PBGs position and behaviors are investigated for different plasma refractive index states and for the applied systematic deformations on layer thicknesses. This deformation follows the function $y = x^{h+1}$, where x and y represent, respectively, the initial non-deformed and the final deformed position (coordinates along the *z*-axis) of the layers, and *h* denotes the coefficient of deformation. It is shown that the PGB spectra exhibit high transmission states, which are very sensitive to the plasma parameters.

## 2. Methodology and Transfer Matrix Formalism

Using the transfer matrix method [27], the electromagnetic responses of 1D deformed and non-deformed quasi-periodic crystals arranged in accordance with the Fibonacci sequence are simulated and theoretically studied. The gallium selenide (GeSe) and plasma materials are used to construct the multilayer Fibonacci structures. We aim to study the sensitivity of the PBGs and the localized mode to the plasma material, the geometric thickness of the layers and the aperiodicity of the Fibonacci photonic structure.

Therefore, the effect of Fibonacci sequence parameters, the deformation of the thickness of the layers using a mathematical rule and the role of the plasma material in the reflectance response of waves will be discussed.

Electromagnetic wave transmission through multilayered stacks is modeled and simulated using the transfer matrix method (also called the Abeles method) [27]. The Abeles method, inspired by Abeles, is adopted to analyze the behavior of electromagnetic wave propagation within the bulk photonic materials. It takes account of the multiple reflectance within the plate. The scattered electromagnetic waves are considered in two forward and backward waves.

We consider a 1D structure consisting of alternating layers of different refractive indices. The amplitudes of the electric fields $E^+$ and $E^-$ of the forward and backward propagating waves and the transmitted wave after m layers $E_{m+1}^+$ are used to calculate the reflectance and the transmittance of waves.

Based on Abeles, the product of these propagation matrices, called the resultant matrix (the transfer matrices for the overall layers of the structures that lie between the amplitudes of the forward and backward propagating waves and transmitted wave after m+1 layers), is given by [28,29]

$$\begin{pmatrix} E_0^+ \\ E_0^- \end{pmatrix} = \prod_{j=1}^{m+1} M_j = \prod_{j=1}^{m+1} \frac{C_j}{t_j} \begin{pmatrix} E_{m+1}^+ \\ E_{m+1}^- \end{pmatrix} \tag{1}$$

where $C_j$ and $t_j$ represent the propagation matrix and the transmittance Fresnel coefficient, respectively.

For the $j$th layer, from the whole structure, the $C_j$ layer's propagation matrix can be written in the form [28,29]

$$C_j = \begin{pmatrix} e^{-i\delta_{j-1}} & r_j e^{-i\delta_{j-1}} \\ r_j e^{i\delta_{j-1}} & e^{i\delta_{j-1}} \end{pmatrix} \tag{2}$$

The transfer matrix of the entire structure can be defined as a product of matrices $\frac{C_j}{t_j}$ [28,29]. Here, $\delta_{j-1}$ represents the phase variation at $(j-1)$th layer $\delta_{j-1} = \frac{2\pi}{\lambda} \hat{n}_{j-1} h_{j-1} \cos\theta_{j-1}$, where $\hat{n}_{j-1}$, $h_{j-1}$ and $\theta_{j-1}$ are, respectively, the refractive indices, the thickness and the incidence angle at the $(j-1)$th layer with $\delta_0 = 0$.

The transfer matrix can be written as [28,29]

$$M = \prod_{j=1}^{m+1} \frac{C_j}{t_j} = \begin{pmatrix} a & b \\ c & d \end{pmatrix} \tag{3}$$

For both TE and TM polarization modes, the transmittance $T$ can be expressed by [28,29]

$$T_s = Re\left(\frac{\hat{n}_{m+1}\cos\theta_{m+1}}{\hat{n}_0\cos\theta_0}\right)|t_s|^2; \ T_p = Re\left(\frac{\hat{n}_{m+1}\cos\theta_{m+1}}{\hat{n}_0\cos\theta_0}\right)|t_p|^2 \tag{4}$$

For both TE and TM polarization modes, the reflectance $R$ can be expressed by [28,29]

$$R_s = |r_s|^2; \ R_p = |r_p|^2 \tag{5}$$

where $r$ and $t$ are, respectively, the reflectance and the transmittance Fresnel coefficient and written as [28,29]

For TE wave polarization (P-mode)

$$t_{jp} = \frac{2\hat{n}_{j-1}\cos\theta_{j-1}}{\hat{n}_{j-1}\cos\theta_j + \hat{n}_j\cos\theta_{j-1}}; \ r_{jp} = \frac{\hat{n}_{j-1}\cos\theta_j - \hat{n}_j\cos\theta_{j-1}}{\hat{n}_{j-1}\cos\theta_j + \hat{n}_j\cos\theta_{j-1}} \tag{6}$$

For TM wave polarization (S-mode)

$$t_{js} = \frac{2\hat{n}_{j-1}\cos\theta_{j-1}}{\hat{n}_{j-1}\cos\theta_{j-1} + \hat{n}_j\cos\theta_j}; \ r_{js} = \frac{\hat{n}_{j-1}\cos\theta_{j-1} - \hat{n}_j\cos\theta_j}{\hat{n}_{j-1}\cos\theta_{j-1} + \hat{n}_j\cos\theta_j} \tag{7}$$

## 3. Plasma Material

Based on Qi et al. [30], we assume that the electron density of the plasma layer, $n_e(z)$, fluctuates sinusoidally along the z-axes. Then, the one-dimensional variation of electronic density of the plasma is written as follows:

$$n_e(z) = n_{e0}[1 - u\sin\left(\frac{m\pi z}{d_p}\right)] \tag{8}$$

Here, $u$, $m$, $d_p$ and $ne_0$ represent the strength (amplitude), the frequency of the sinusoidal variation, the plasma layer thickness and the electron density for the uniform plasma case, respectively, of the plasma sinusoidal variation. In a cold non-magnetized plasma, the dielectric permittivity and the plasma frequency can be written as [30]

$$\epsilon(z) = 1 - \frac{\omega_p^2(z)}{\omega(\omega + i\gamma)}; \; \omega_p^2(z) = \frac{4\pi e^2 n_e(z)}{m_e} \tag{9}$$

where $m_e = 9.1 \times 10^{-31}$ kg, $e = 1.6 \times 10^{-19} C$, and $\gamma$ denotes the electron mass, the electron charge and the collision frequency, respectively [30]. Based on Equations (8) and (9), dielectric permittivity can be expressed as [30]

$$\epsilon(z) = \left(1 - \frac{\Omega^2}{\omega(\omega + i\gamma)}\right) + \frac{\Omega^2}{\omega(\omega + i\gamma)}\sin(\frac{m\pi z}{d_p}) \tag{10}$$

where $\Omega^2 = \frac{4\pi e^2 n_{e0} e^2 u}{m}$.

In the first order approximation, the refractive index of the modulated sinusoidal plasma layer is given by [30,31]

$$n(z) = n_0 + a\sin(\frac{m\pi z}{d_p}) \tag{11}$$

where the factors $n_0$ and $a$ are frequency-dependent and given by the plasma parameters [30,31]

$$n_0 = (1 - \frac{\Omega^2}{\omega(\omega + i\gamma)})^{\frac{1}{2}} \; \text{ and } a = \frac{\Omega^2}{2[\omega(\omega + i\gamma) - \Omega^2]} \tag{12}$$

In the simulation, we determine that $n_0 = 0.5$ and $a$ are frequency-independent (low-density plasma type with collision-less particles, $\gamma = 0$). The thickness of the plasma layer is set to be $d_p = 10$ dB.

## 4. Results and Discussion

In this paper, the electromagnetic behavior of quasi-periodic selective (multichannel band gaps\multi-Bragg peaks) photonic structures is theoretically studied at THz frequencies. Our objective is to tune the position and the number of the resonance peaks, as well the position and the width of the PBGs. These tunable properties open the way to applying the proposed structures as filters for the communication field, sensors or micro-cavities for LASER equipment. The photonic structures (Figure 1) are made of a GaSe semiconductor material (with an indirect band gap) and sinusoidal plasma material (with permittivity $\varepsilon_p$). If this study is experimentally validated in the future, we intend to insert two transparent polymer film materials at the top and bottom of the structure (two red lines in Figure 1), which are intended to protect the plasma material and prevent it from escaping and spreading outside the structure. This polymer material should be transparent, and we assume that that has no effect on the propagation of waves. The most transparent and commercial polymers are Polycarbonate, PMMA or Acrylic, Polyethylene Terephthalate (PET), Amorphous Copolyester (PETG), Polyvinyl Chloride (PVC), Liquid Silicone Rubber (LSR), Cyclic Olefin Copolymers (COC), Polyethylene (PE), Ionomer Resin, Transparent Polypropylene (PP), Fluorinated Ethylene Propylene (FEP), Styrene Methyl Methacrylate (SMMA), Styrene Acrylonitrile Resin (SAN), Polystyrene (General Purpose—GPPS) and MABS (Transparent ABS) materials. The proposed structure is designed to confine the light at different resonant frequencies. The PC lattice is built according to the Fibonacci sequence, where the multilayered stacks are recursively generated by the substitution rule $\sigma_F(H, L) : H \rightarrow HL$ , $L \rightarrow H$ ([21,25]). According to the recursive substitution rule above, the fractal Fibonacci structure $S_{k+1}$ can be written as $S_{l+1} = S_l S_{l-1}$, where $l$ is the generation number of the Fibonacci sequence.

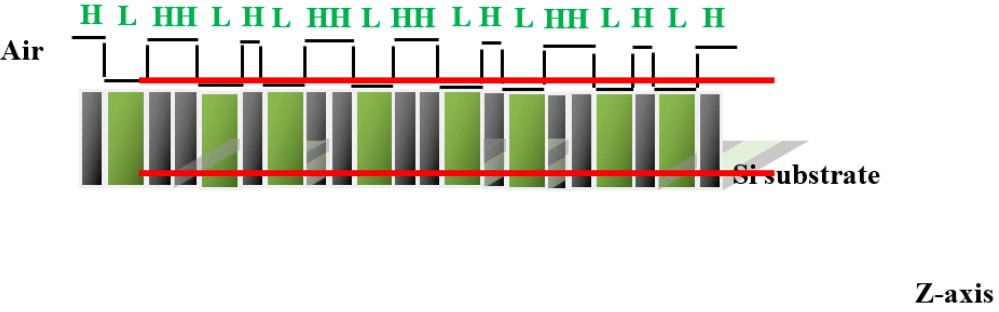

**Figure 1.** A schematic representation of the proposed multilayer PC structure according to the seventh generation of the classic Fibonacci sequence. The multilayers are made of GaSe (layers H) and sinusoidal plasma (layers L) materials.

In the input of the PC (the left side of the structure), we consider an incident plane wave coming at an angle $\theta$ from the z direction. The refractive indices of the boundary limits are equal to 1 (Air). The plasma (layers L) and GaSe (layers H) materials have refractive indices $n_P$ and $n_H = 2.43$, respectively. The plasma-dependent refractive index is given in Equation (10). The electromagnetic properties (such as reflectance and electric field) of the tunable photonic band gaps (PBGs) were determined and simulated using the transfer matrix method (TMM).

In this part, the effect of the iteration number ($l$) of the Fibonacci sequence on the reflectance spectrum is studied. Figure 2 shows the reflectance spectra for different iteration numbers ($l$) of the Fibonacci (F(1,1)) sequence. In this step, the thickness of the layers is still not deformed ($h = 0$), and the incidence angle of wave $\theta$ varies from 0 to 90 degrees. The classic Fibonacci sequence F (1, 1) exhibits a multichannel sub-band gap at different values of the iteration number. These relevant gaps are due to the Bragg scattering. In Figure 2, we notice that the bandwidth of each PBG decreases with the increase in the Fibonacci order ($l$), whereas the number of gaps increases according to this order. This is due to the confinement of light within the stacks introduced by the increase in the number of cavities (local defects) when increasing the number of layers (due to the increase in the Fibonacci sequence) of the whole structure. Therefore, the auto-similarity of defects increases with the Fibonacci order. Here, and compared with the conventional periodic structures, it is clear that the increase in the fractal order of the quasi-periodic structure has a significant effect on the bulk material properties and the transmission of waves. In addition, a multi-oscillation beam with a different resonance mode was found between the PBGs when the incidence wave angle was $\theta \prec \frac{\pi}{2}$.

In this part, we study the effect of the deformation of layer thicknesses on the position of the PBGs and the intensity of the resonance peaks. By modeling the deposition layer by layer on the silicon substrates of the main plasma quasi-periodic PCs (PQPCs), we assume that the deformation of layer thicknesses approximately follows the mathematic law $y = x^{h+1}$, where $x$ and $y$ represent the main and the deformed z-coordinate of the layers, respectively, and $h$ denotes the deformation degree. Figure 3 shows the reflectance spectra of PQPCs as a function of frequency (Hz) and the incidence angle ($\theta$) and for different values of the deformation degree $h$. It is clear that the reflectance of waves is sensitive to the applied deformation strain. By increasing the deformation degree $h$, the intensity of the output beam when $h = 0$ (see Figure 2) decreases progressively, and several pseudo-PBGs appear, where the propagation of waves is inhibited. The actual PBGs shift toward the low frequencies, and an enlargement of their width is noted. Physically, the increase in the deformation strain allows an increase in the geometric thicknesses of the layers, which corresponds to the adjustment of the refractive indices' contrast. This phenomenon influences the propagation of waves and the distribution of the electromagnetic field. Here, in Figure 2, we note that the maximum shift of PBGs is obtained when $h = 0.02$. In addition, the majority of the resonance peaks are attenuated.

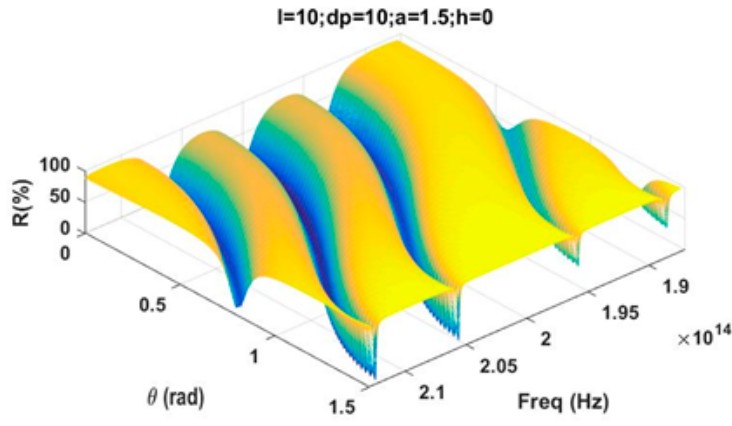

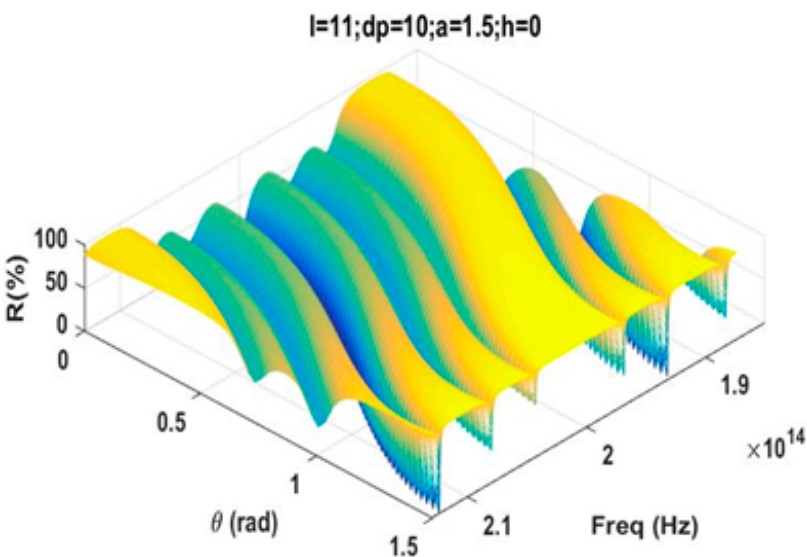

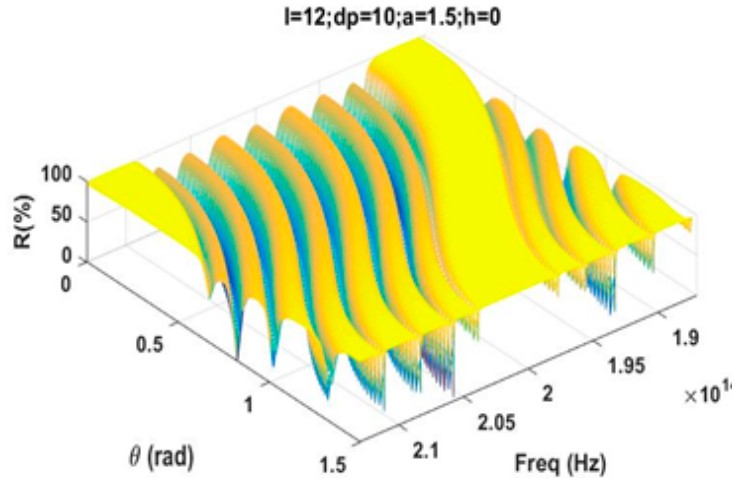

**Figure 2.** *Cont.*

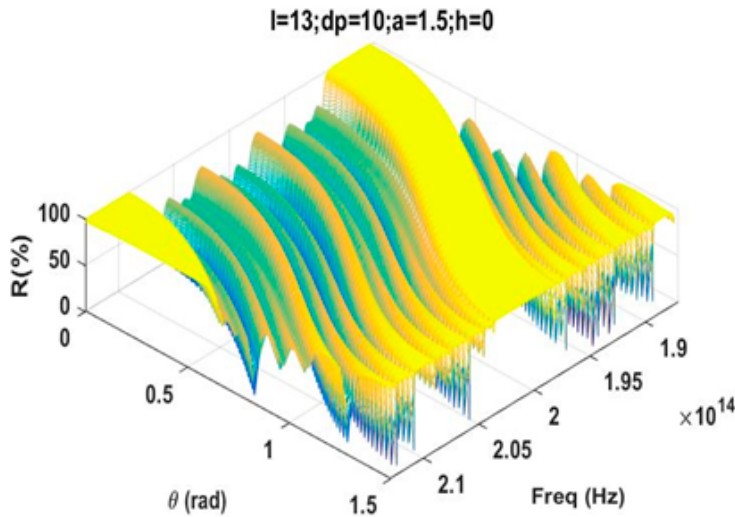

**Figure 2.** Reflectance spectra of quasi-periodic Fibonacci multilayered stacks as function of frequency and the incidence angle ($\theta$) and for different sequence orders (*l* set to be from 10 to 13).

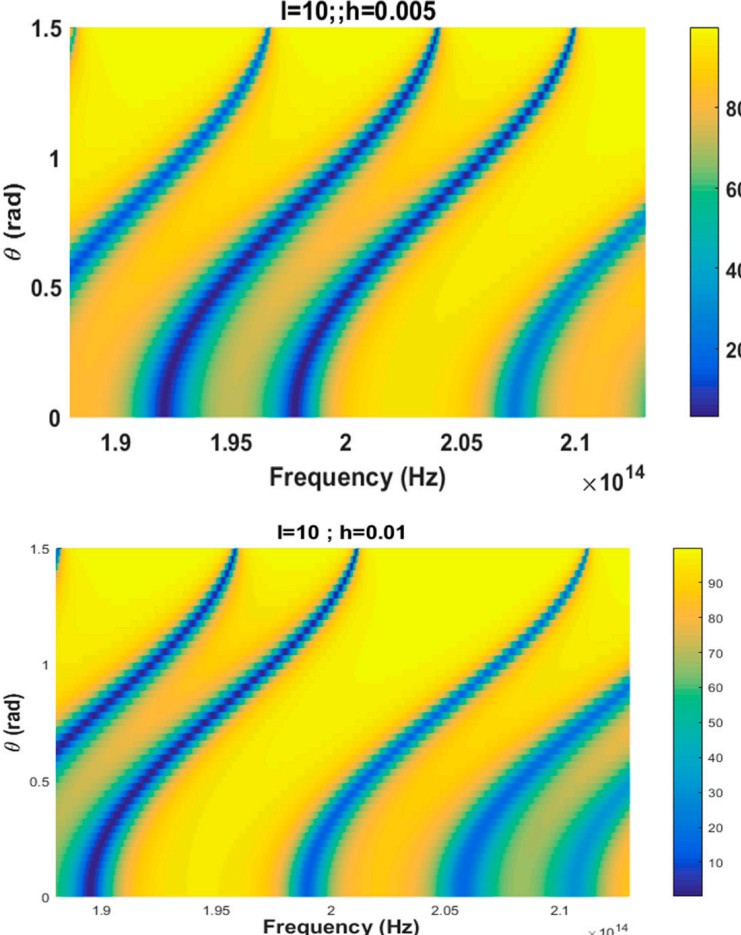

**Figure 3.** *Cont.*

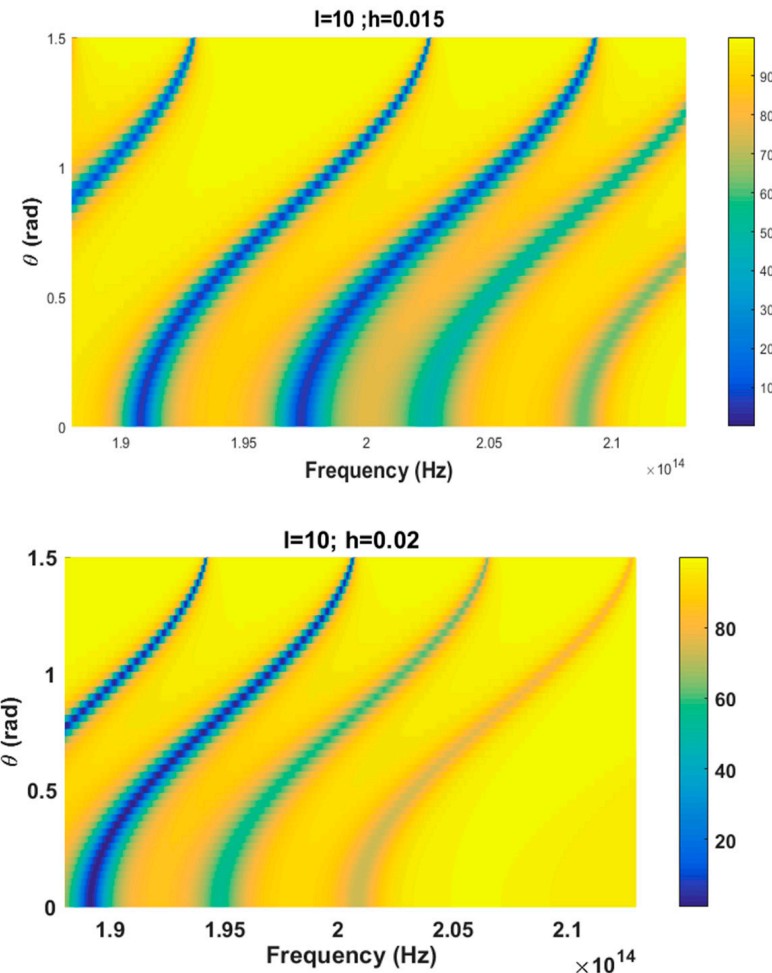

**Figure 3.** Reflectance spectra of deformed quasi-periodic Fibonacci multilayered stacks as function of frequency and the incidence angle ($\theta$) and for different deformation degrees ($h$ set to be from 0.005 to 0.02).

In Equation (11), we see that the refractive index of the modulated sinusoidal plasma layer depends on two parameters, $m$ and $a$. In this part, we study the effect of the variation of the coefficient $m$ of sinusoidal plasma layers on the reflectance spectra, and in the next part, we will study the effect of parameter $a$. In Figure 4, we note that by changing the coefficient $m$, the output beam intensity can be controlled. The oscillation beams in the reflectance spectrum appear in the blue zones between the PBGs. When $m = 10$, the intensity of these beams is maximal; afterward, and when parameter $m$ is changed to 20, the intensity of these beams attenuates, and again, they reappear with high intensity when $m = 50$ (see the blue zones with red circles in Figure 4). The presence of these oscillation beams is due to the sinusoidal refractive index formula of the plasma layers. Therefore, the modulation of the plasma refractive index permits the modulation of the resonance peaks between PBGs. However, the tuning of parameter $m$ has no effect on the position of the PBGs. These modulations are due to the Bragg resonance phenomena occurring in the building blocks, which show a weaker modulation and a narrower range of transmitted frequencies.

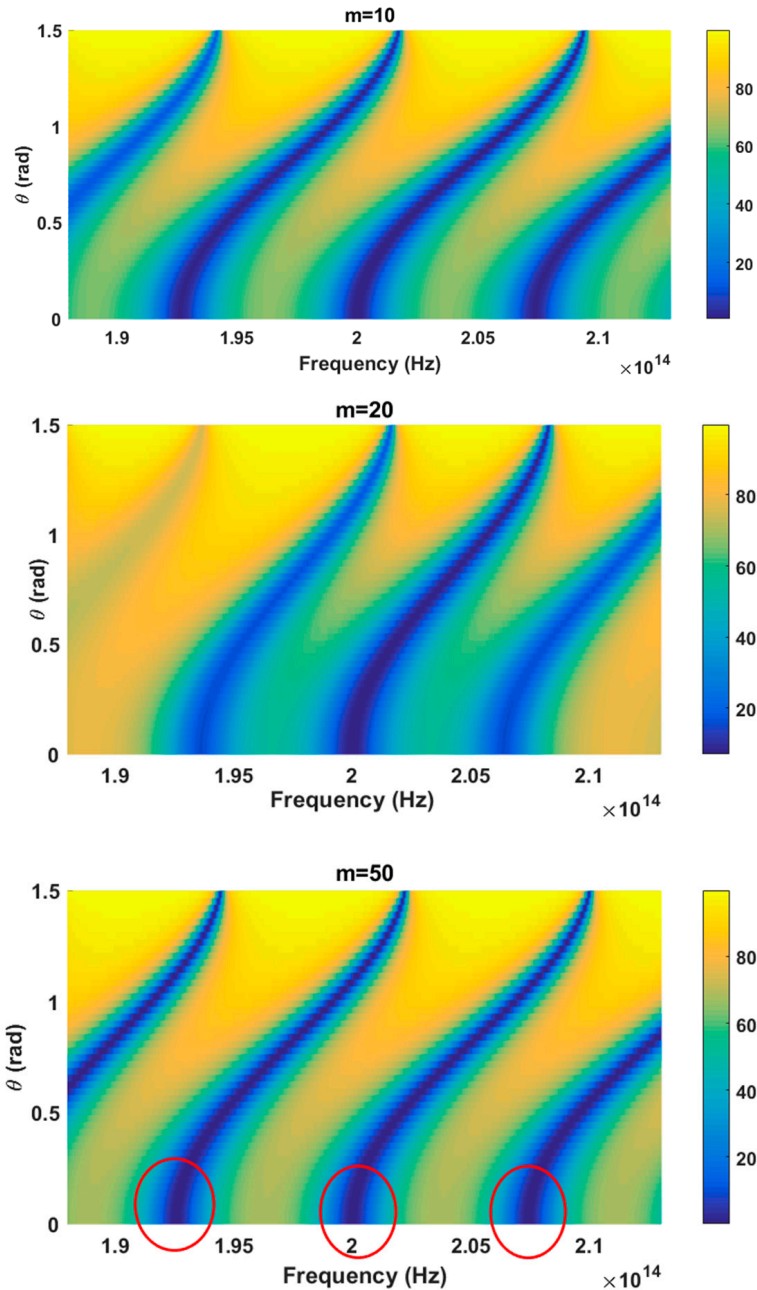

**Figure 4.** Reflectance spectra of quasi-periodic Fibonacci multilayered stacks as function of frequency and the incidence angle (*θ*) and for different values of parameter *m* of the plasma refractive index (*m* set to be 10, 20 and 50; *l* = 10; and *dp* = 10).

Figure 5 shows the reflectance spectra properties when parameter *a* of the plasma layer refractive index varies from 2 to 10. By increasing parameter *a*, the PBGs disappear. In addition, the intensity of the output beams decreases with *a*. The resonance peak positions remain at the same frequencies $f_1 = 193$ THz, $f_2 = 200$ THz and $f_3 = 207$ THz. Therefore, the variation in parameter *a* only enables hiding the PBGs and controlling the intensity of the resonance peaks.

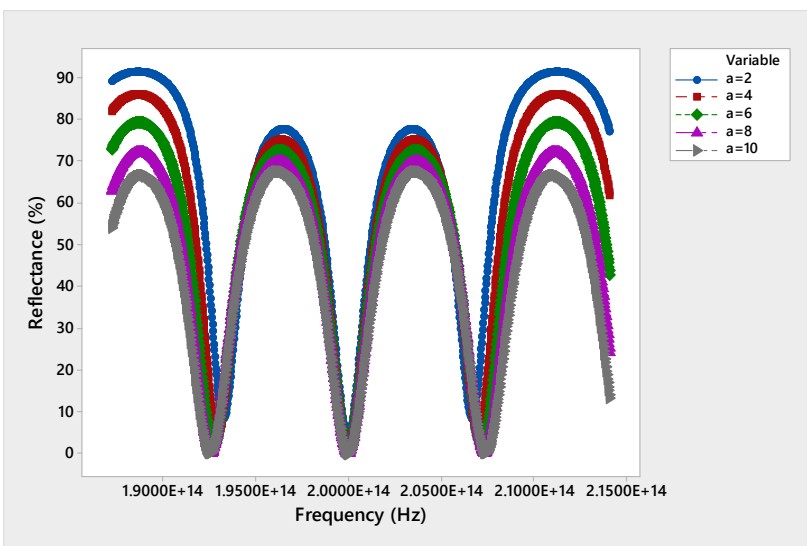

**Figure 5.** Reflectance spectra of quasi-periodic Fibonacci multilayered stacks as function of frequency and for different values of parameter *a* of the plasma refractive index (*a* set to be 2, 4, 6, 8 and 10; *l* = 10; and *dp* = 10).

As in other previously published scientific works concerning the management of the position, number and width of PBGs using the Fibonacci photonic structure, this current work adds another property with the use of the plasma material. In Figures 4 and 5, and in comparison with previously published papers, the advantage of using the plasma material in a Fibonacci structure instead of ordinary semiconductor [32] and superconductor [33] materials is clear, enabling the possibility of tuning the intensity of the resonance peaks between PBGs, as well as the possibility of hiding (or showing) the PBGs. These electromagnetic behaviors of our structures have the potential to withstand high-power microwaves and pave the way for their use in filtering applications for communication and as micro-cavities to localize waves and to develop intense LASER equipment [34].

## 5. Conclusions

Using the TMM method, the beam wave propagation along 1D quasi-periodic GeSe/plasma crystal is theoretically studied. The quasi-periodicity is built according to the classic Fibonacci sequence. In the first step of this study, the behavior of the wave beam appearing in the reflectance spectra is analyzed for different orders of the Fibonacci sequence. Afterward, we apply a deformation mathematical law $y = x^{h+1}$ to the layer thicknesses of the strucutre, where $x$ and $y$ represent the main and the deformed z-coordinates of the layers, respectively, and $h$ denotes the deformation degree. By increasing the deformation degree $h$, the intensity of the output beam decreases progressively, and several pseudo-PBGs appear, where the propagation of waves is inhibited. The actual PBGs shift toward the low frequencies, and an enlargement of their width is noted. In addition, the majority of the resonance peak intensities are attenuated. In the third step, the effect of parameter $m$ on the refractive index formula of the plasma material is studied. When $m$ = 10, the intensity of the output beams is maximal; afterward, and when parameter $m$ increases to 20, the intensity of these beams attenuates, and again, they reappear with high intensity when $m$ reaches 50. In the fourth step, the effect of parameter $a$ on the refractive index formula of the plasma material is studied. The variation in $a$ enables hiding the PBGs and controlling the intensity of the resonance peaks, and for different values of $a$, we notice that these peaks are still in the same positions ($f_1 = 193$ THZ, $f_2 = 200$ THZ and $f_3 = 207$ THZ). These results show that the proposed photonic system can be driven by the states of the parameters of the involved plasma material, as well as the iteration and the thickness of the whole layers. These electromagnetic behaviors of the theoretically studied photonic systems open the

way to experimentally validating these simulated results in the future. These structures have the potential to withstand high-power microwaves and pave the way for their use in filtering applications for communication and as micro-cavities to localize waves and to develop intense LASER equipment [34].

**Author Contributions:** Conceptualization, N.B.A. and Y.T.; methodology, H.A. and N.B.A.; software, N.B.A. H.A., O.K. and Z.E.; validation, N.B.A., Y.T. and H.A.; formal analysis, N.B.A. H.A., O.K. and Z.E.; investigation, Y.T. and H.A.; resources, Y.T., N.B.A. and H.A.; writing—original draft preparation, all authors.; writing—review and editing, Y.T.; visualization, N.B.A., O.K. and Z.E.; supervision, N.B.A.; project administration, funding acquisition, all authors. All authors have read and agreed to the published version of the manuscript.

**Funding:** This research has been funded by Scientific Research Deanship at University of Ha'il—Saudi Arabia through project number <<RG-23 031>>.

**Institutional Review Board Statement:** Not applicable.

**Informed Consent Statement:** Not applicable.

**Data Availability Statement:** The data will be made available upon reasonable request from the corresponding author.

**Conflicts of Interest:** The authors declare no conflict of interest.

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
