# Peer review of "Localized Modes and Photonic Band Gap Sensitivities with 1D Fibonacci Quasi-Crystals Filled with Sinusoidal Modulated Plasma"

_applsci, doi:10.3390/app13158641_

Round 1

Reviewer 1 Report

The manuscript “Localized modes and Photonic band gap sensitivities with a sinusoidal modulated plasma filled in 1D Fibonacci quasicrystals” by Naim Ben Ali, Youssef Trabelsi, Haitham Alsaif, Omar Kahouli and Zied Elleuch deals with the spectral characteristics of a one-dimensional photonic crystal (PC), which is characterized by a non-periodic alternation of layers and the use of “plasma material”. It is shown how different parameters of the structure and materials affect the reflection spectrum of the PC. It has been found that varying the parameters of the plasmonic material makes it possible to modify the photonic band gap of the PC. Apparently, the originality and novelty of this work lies in the study of the photonic structure, which simultaneously uses the non-periodicity of the alternation of layers and the plasmon material. In general, the level of studies carried out and the results obtained are moderate. The authors do not explain what a plasmonic material is and what optical properties it has. Many parameters are not given in the manuscript, including the values of n_e, d_p, gamma, u, the frequency of plasma layer, the thickness of GaSe layers. The value of the GaSe index (nH = 3.26) used in the calculations does not correspond to the value of the index near the considered frequency range (nH = 2.43 – see, for example, https://refractiveindex.info/?shelf=main&book=GaSe&page=Kato-e). I cannot recommend publishing this paper in its current form in view of serious critical remarks given above.

I recommend authors to bring work to improve the language quality, as well as correct the following shortcomings:

- line 102: r and t are coefficient of reflection and transmission (not “the Fresnel equations”)

- lines 100-106: Equations (4)-(5) include unreadable characters.

- Need to explain what is Fibonacci order (l), and t_j in Eq. (1).

- line 76-81: Remove the repetition of the reference to article [24] at the end of each sentence.

- line 59-61: Add a reference to the paper that outlines this study.

- introduction: In the introduction part, the authors can do improvements by adding a few up-to-date pieces of information about the aperiodic PCs, for example https://doi.org/10.1063/5.0008652 and https://doi.org/10.1364/OL.445779. This will make your introduction more relevant and interesting.

Author Response

Please find attached the response to the editor and reviewers' comments.

Thank you

Reviewer 2 Report

Dear Authors,

The research direction of this submitted manuscript is intriguing. However, there are several areas that need improvement. Firstly, the introduction section lacks clarity in expressing the research objectives and significance. It would be beneficial to provide a more comprehensive and concise introduction to better engage readers and highlight the importance of the study. 

Secondly, the methodology section requires more detailed explanations and clear descriptions of the experimental design. It is important to provide sufficient information for readers to understand the procedures and techniques used in the study.

Furthermore, there are issues with the presentation of figures and the discussion of results. There are numerous spelling and numerical errors throughout the manuscript that need to be addressed. It is crucial to thoroughly proofread and revise the manuscript to ensure accuracy and coherence.

Overall, I encourage you to revise and reorganize the manuscript, ensuring a clear and concise presentation of the research findings. Pay close attention to grammar, spelling, and clarity of expression throughout the manuscript.

Thank you for considering these suggestions, and I look forward to seeing the revised version of this interesting research.

Best regards,

no comments. thanks

Author Response

(The authors gave the same response as above.)

Round 2

Reviewer 1 Report

The following basic issues must be resolved by the authors so that it can be considered for publication in such a reputed journal:

The manuscript does not provide the values of the physical quantities n_e, d_p, gamma, u, and others. What natural material are the "plasma layers" made of? How can the parameters m and a of the plasma refractive index be varied (m set to be 10, 20 and 50, and a from 2 to 10)?

Minor editing recommendation. Authors are advised to check the style of new references [17], [18] and [26] carefully.
